# Measuring Comprehensive Production Efficiency of the Chinese Construction Industry: A Bootstrap-DEA-Malmquist Approach

**Aobo Yue [1] and Xupeng Yin [2,*]**

1   School of Management Science and Real Estate, Chongqing University, Chongqing 400044, China
2   School of Management, Henan University of Urban Construction, Pingdingshan 467036, China
*   Correspondence: yin1007hncj@163.com; Tel.: +86-17347816969

**Abstract:** Production efficiency is a critical research topic in the field of construction economics and management. It reflects the developmental potential and competitiveness of the economy or an economic system. An objective and reasonable assessment framework of the production efficiency in the construction industry is essential to promote the industry's high-quality development. This study aims to propose a scientific and holistic framework to examine the production efficiency in the construction industry and to investigate evolution patterns from a macroeconomic perspective. Input and output indicators were identified through the value-added and the fuzzy Delphi methods. In addition, the production efficiency in the construction industry was examined via the bootstrap-DEA and Malmquist exponential decomposition models. A case study in China was conducted at the end of this research. The panel data of 31 provinces from 2010 to 2020 were applied in the case study. The results reveal the following: (1) The bootstrap-DEA model results show that the trends of production efficiency before and after rectification are similar, but the difference is largest at the peak. Moreover, the production efficiency value after correction is evidently lower than that which is obtained by the traditional DEA model. (2) The Malmquist index decomposition results show that the change trend of technical efficiency in the construction industry is contrary to that of the scale efficiency. In addition, the improvement of scale efficiency cannot bring a melioration of management efficiency or the accumulation of production experience. (3) There is no direct correlation between production efficiency and economic development. High-value areas and median areas are contiguous, and they are mainly distributed in the central and eastern provinces. The findings accurately reflect construction industry productivity, providing practical data for developing policy recommendations for bridging regional construction development gaps.

**Keywords:** production efficiency; construction industry; bootstrap-DEA-Malmquist; comprehensive evaluation; regional differences

## 1. Introduction

The construction industry has made considerable contributions to the development of the world economy through its multiplier effects. These contributions include providing housing and infrastructure, increasing employment, and boosting domestic consumption, especially in countries experiencing rapid urbanization, such as China. China may have the largest construction market in the world [1]. The construction industry in China is a pillar industry. It not only improves the built environment, but also boosts the economy and provides employment opportunities [2,3]. Therefore, investigating the productivity and efficiency of China's construction industry is crucial to attract potential foreign investors and practitioners.

Productivity, which is a significant index of production performance, is to measure the ability of a production unit, aiming at achieving maximum outputs by using the available technologies [4,5]. Government policy is an essential factor that affects the economy. Productivity analysis provides valuable information on the effectiveness of

economic policies, which is a useful tool for policy design. Currently, China is undergoing an economic transformation. In the current context, productivity is more important than other efficiency indicators. Unfortunately, the productivity of the construction industry has not received much attention for a long time, that is, until the publication of Professor David Pearce's report "The Social and Economic Value of the Construction Industry" [6], which is when the efficiency of the construction industry began to attract the attention of academics. However, compared with other industries, there is still less attention paid to research regarding the construction industry.

China is a vast country with unbalanced economic development between regions, and the development of the construction industry shows large differences. The long-term existence and excessive expansion of such differences not only affect the overall efficiency of the construction industry, but also influence the effective allocation of resources [7–9]. Therefore, narrowing the efficiency gap between regions and improving regional coordination should become the focus of future research, and analyzing the differences and convergence of the production efficiency of the regional construction industry is the key issue that needs to be resolved in this field. A review of the literature on measuring productivity in the Chinese construction industry shows that most studies use data envelopment analysis (DEA) to estimate productivity. DEA uses a nonparametric linear programming model, which is important for use in real-world conditions to measure the performance of decision-making units (DMUs). Behind the advantages of the original DEA model, there are some disadvantages of the current research: (1) Conventional DEA with data envelopment analysis suffers from incomplete selection of indicators as well as lack of rationality, which will directly affect the reliability of the evaluation results; (2) conventional DEA models are prone to statistical errors, omitted data variables, and other random shocks, which seem to overestimate DMU evaluation efficiency and fail to reflect the true; (3) conventional DEA models are mainly used for static evaluation, and research on dynamic evaluation is only based on the static evaluation results, considering the simple comparison of different time series cross sections, but rarely includes the analysis of the characteristics of the speed of change within a certain time series interval, so it cannot fully reflect the overall speed change development trend of the evaluation object. To solve this problem, scholars have tried to provide hybrid methods to improve the capability of DEA under different conditions. For example, to improve the rationality of indicator system construction, scholars have tried to use DEA-OPA [10], DEA-AHP [11], DEA-Delphi [12], and other methods to improve the rationality of indicator selection with the help of experts' empirical knowledge. However, the problem of a single assessment dimension is not yet overcome by these methods. To overcome it, scholars have introduced the Malmquist-DEA model to perform dynamic evaluation of evaluation objects. Some scholars have also combined static DEA models with dynamic DEA models to conduct efficiency evaluations in the financial sector. However, to the best of the authors' knowledge, no article has been published on the application of a combination of static and dynamic DEA methods to the evaluation of productivity in the construction industry. In other words, in the area of productivity assessment in the construction industry, the combination of static efficiency evaluation and dynamic efficiency evaluation has not yet been effective, and the evaluation of the overall construction industry production efficiency is not comprehensive enough, such that there is a need to continuously improve the construction industry production efficiency assessment methods. Based on this, the aim of this study is to define a comprehensive indicator for assessing the productivity of the construction industry in the Chinese provinces, and the method used can improve the scientific approach to measuring the productivity of the construction industry. Therefore, the contributions of the current study are manifold and can be expressed as follows: (1) A model combining the value-added method and the fuzzy Delphi method is proposed. The model can strengthen the purpose of productivity evaluation based on the measurement method and improve the coverage of the indicator system by combining expert empirical knowledge to increase the rationality of the indicator system. (2) A static DEA evaluation model based on the bootstrap technique is proposed.

The bootstrap technique based on repeated self-sampling can eliminate the interference of random factors and the errors caused by omitted variables, and achieve a more accurate measurement of the estimated efficiency and its changes in the traditional DEA model. (3) An evaluation idea of the comprehensive production efficiency of China's construction industry based on a bootstrap DEA and Malmquist index decomposition model is proposed. This research idea achieves an integrated study on the selection of production efficiency evaluation indicators and the establishment of evaluation system and model metrics, which avoids the influence of subjective factors and error factors to the maximum extent, and provides a scientific quantitative evaluation method for the construction of a production efficiency evaluation index system.

The rest of this research is organized as follows: Section 2 highlights a literature review on previous studies. The research methodology is presented in Section 3. The research results and an in-depth discussion are presented in Sections 4 and 5, respectively, whereas Section 6 details the research conclusion and provides guidance on future research directions.

## 2. Literature Review

### 2.1. Productivity in the Construction Industry

There are many studies on productivity in the construction industry, especially related to the factors and trends affecting the industry's productivity [13–15]. These studies help decision makers to determine the overall efficiency trends of production units. There are two types of mainstream methods for conducting efficiency assessments in the current study: parametric and nonparametric methods. The parametric method is represented by stochastic frontier analysis (SFA), while the nonparametric method is represented by data envelopment analysis (DEA). When focusing on the production efficiency of the construction industry, from the perspective of research methods, there are three methods used in production efficiency research by scholars: (1) The production efficiency of the construction industry was evaluated by the C–D production function [16], the traditional DEA model [9,17], and the multistage or extended DEA model [18,19]. Zhu et al. (2019) used three-staged DEA models to analyze the impact of environmental regulations on the regional construction industry's productivity, based on panel data from 30 provinces in China from 2011 to 2015 [16]. (2) The DEA-Malmquist exponential decomposition model was used to dynamically evaluate the changes in production efficiency of the construction industry. Xu et al. (2019) measured the change in energy productivity in the construction industry from 2004 to 2009 using input-oriented models with data from 26 Chinese provinces [9]. Chen et al. (2019) aimed to measure the evolution of the de-stocking performance of the Chinese real estate industry based on a DEA-Malmquist approach [13]. Nazarko and Chodakowska (2015) used the DEA method to calculate labor productivity. Further, the change in efficiency over the period of 2006–2012 was estimated through the Malmquist index [20]. Tobit regression was also applied to explore the impact of a country's economic performance on its labor productivity in the construction industry.

In terms of the study population, the method has been widely used to assess input–output efficiency in several sectors [21], such as electricity [22], agriculture [23], manufacturing [24], and infrastructure [25]. In the construction sector, DEA has been conducted from both environmental and economic perspectives. The methodology has been used to assess industrial sustainability [26], energy efficiency [27], and carbon emissions [28] from the environmental perspective. Several studies have measured the economic performance of construction projects from an economic perspective [29,30], and the construction performance [31] from an economic perspective.

### 2.2. Indicator Selection

Carrying out an evaluation study of productivity is systematic and long-term work. Previous studies have shown that the selection of input–output indicators is the key factor that affects the evaluation results. This study summarizes the input–output indicators in

the construction industry based on an extensive literature review on the efficiency of the construction industry, as shown in Table 1. Zhang et al. (2018) used a three-staged DEA model to identify the efficiency input index (total wages of construction workers, number of engaged persons, total assets, total power of machinery, and equipment owned) and output index (engineering settlement profits, floor space of building, gross output value, and total profits of China's construction industry) from 2011 to 2015 [15]. Hu and Liu (2016) constructed an evaluation system with the gross value added as input indicators and the gross operating surplus and mixed income as the output indicators [32]. The productivity of the Australian construction industry was evaluated based on a two-stage DEA model. Wang et al. (2020) used the Solow residual approach to evaluate the efficiency of the construction industry in China, wherein fixed assets and the number of employees served as the input index, and the total value added was understood as the output index [5]. Based on the super efficiency DEA model and the ANN model, Yuan et al. (2020) studied the efficiency of China's construction industry from 2000 to 2017, with the number of employed persons, total assets, total capacity of machinery, and equipment owned as the input index and the gross product of the construction industry and newly built floor area as the output index [14].

### 2.3. Research Gaps

In summary, the existing research mainly has the following limitations: (1) DEA technology has been widely applied in the construction industry, but it has several drawbacks. First, most studies were focused on holistic economic performance within the economic system, while regional differences were ignored. There are imbalances in the distribution of construction resources and economies across China, leading to regional differences in productivity and technical efficiency. These regional differences have an indirect but important impact on the local construction sector, which may lead to misconceptions about the actual performance of these sectors. (2) The static evaluation method measurement results can only be used as an efficiency value at a certain time, and the measurement, without considering the time dimension, largely limits the fairness and objectivity of the efficiency evaluation. Most particularly under the context of reform in the construction industry, the efficiency evaluation without the time dimension can easily mislead decision makers. They may abandon the long-term development of the industry for high performance at a certain point in time. Although the dynamic evaluation method makes up for this defect, the results obtained by the measurement are all indicators of the change rate, and it is difficult to serve as a variable analyzed in the regression model directly. (3) There have been more applications of DEA methods in the related literature, but the traditional DEA measurement only gives point estimates without considering the influence of random factors, which has obvious shortcomings: it is difficult to avoid the problems of sample sensitivity and the influence of extreme values. The bootstrap method can enlarge the sample size by repeated sampling, which brings the sample size closer to the overall, and reduces the statistical errors brought about by small samples [21]. (4) It is clear that the DEA is the dominant method for studying productivity in the Chinese construction sector, and that the main area of inconsistency is in the choice of inputs and outputs. Therefore, the scientific selection of input and output indicators will be a key step in the study. Similarly, previous studies have mainly focused on a single perspective of static performance or dynamic change, which has failed to fully reflect the level and potential of productivity development in the construction industry and, to some extent, has hindered the comprehensive evaluation of productivity development in the construction industry in different regions of China and the proposal of corresponding optimization strategies. This study aims to propose a scientific and holistic framework to study the productivity of the construction industry and to examine the development patterns from a macroeconomic perspective. The value-added method and the fuzzy Delphi method are used to determine the input and output indicators, and the bootstrap DEA and Malmquist index decomposition models are used to examine the productivity of the construction industry.

**Table 1.** Input and output variables of the construction industry obtained from the literature review.

| References | Research Objects | Input Indicators | Output Indicators | Methods |
|---|---|---|---|---|
| Zhang et al. (2018) [15] | China's construction industry | Total wages of construction workers, number of engaged persons, total assets, total power of machinery, and equipment owned | Engineering settlement profits, floor space of building, gross output value, and total profits | 3-stage DEA method |
| Hu and Liu (2016) [32] | Australian construction industry | Gross value added | Gross operating surplus and mixed income | 2-stage DEA method |
| Chancellor and Lu (2016) [33] | China's construction industry | Number of construction workers and staff at year end, paid-up total capital, total assets, total power of machinery, and equipment owned | Total floor space of buildings completed and total output value of construction | DEA |
| Wang et al. (2020) [5] | China's construction industry | Fixed assets and number of employees | Total value added | Solow residual approach |
| Yuan et al. (2020) [14] | China's construction industry | Number of employed persons, total assets, total capacity of machinery, and equipment owned | Gross product of the construction industry and newly built floor area | Super-efficiency-DEA |
| Yang et al. (2019) [26] | China's construction industry | Built-up area, total number of employees, capital stock, energy consumption, and total water usage | Industrial solid wastes produced, industrial waste gas emissions, gross domestic product, et al. | DEA and DDFs |
| Huo et al. (2018) [34] | China's construction industry | Labor force, total assets of construction enterprises, total capacity of machinery, equipment owned, and energy | Gross output value in the construction industry and floor space of buildings under construction | Luenberger productivity index and DDFs |
| Li and Song (2012) [35] | China's construction industry | Labor force and assets of construction enterprises | Value added and total solid waste | Malmquist–Luenberger |
| Tong et al. (2022) [36] | China's construction industry | Capital, labor, energy, machinery, material | Total output value, total pretax profit, and floor space of buildings, undesirable environmental outputs | Windows-Super-SBM model |
| Li et al. (2021) [37] | US construction industry | Number of workers and managers per year for each state | GDP of the construction industry | DEA-Malmquist |
| Chen et al. (2021) [38] | China's construction industry | Capital, labor, energy, material | Gross output value of construction, total profits, and completed floor area, undesirable environmental outputs | 3-stage SBM-DEA model |

## 3. Research Methodology

### 3.1. Research Framework

In this study, two research questions should be addressed: (1) how to reasonably build the input–output index system to evaluate the production efficiency of the construction industry, and (2) how to improve the traditional evaluation model in order to obtain the objective value of the production efficiency of the construction industry. Therefore, in order to achieve the objectives of this study, as shown in Figure 1, the research framework can be divided into three different stages: index sorting and establishing an evaluation index system, the rectification of static production efficiency, and a solution of the comprehensive production efficiency.

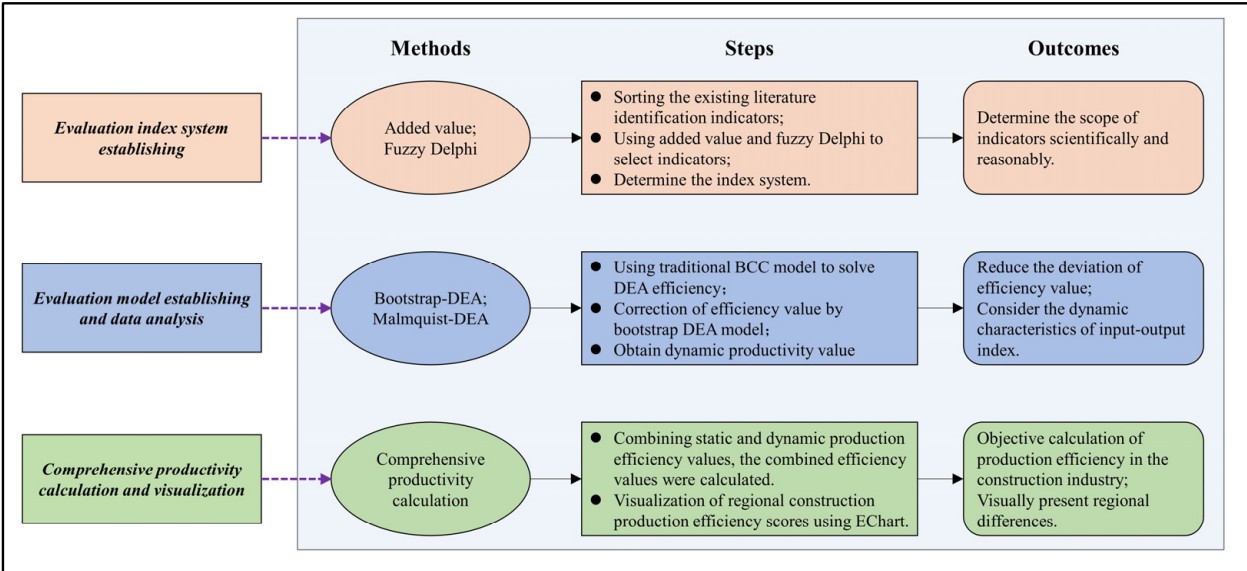

**Figure 1.** Research framework.

### 3.1.1. Evaluation Index System Establishment

Step 1 is to establish an evaluation index system. This paper sorts and summarizes the common indicators via an extensive literature review. Then, it establishes a construction industry productivity evaluation index system with the help of the value-added method and Delphi method in order to increase the credibility and rationality of the evaluation results [39]. Conducting research on the evaluation of production efficiency is systematic and long-term work. Previous studies have shown that the choice of input–output indicators is a key factor in the evaluation results. However, research on the evaluation of production efficiency in the construction industry is directly based on experience. Liu et al. (2013) combined factor analysis with correlation analysis to identify the factors, making the selection of indicators more scientific [39]. However, there are still certain shortcomings in this process; that is, the factor identification by factor loadings may filter out some indicators with small weights but not large values. Furthermore, this phenomenon may cause bias in the model calculation. The quantitative research method can be used to reduce the index, but it is contrary to the DEA model principle. Therefore, it is not enough to rely solely on complex measurement models for the construction of input–output evaluation indicators. Empirical research should also be based on the connotation of an empirical basis, the use of measurement methods to strengthen the purpose of evaluation, and relying on empirical analysis to determine the evaluation. Accordingly, this study used the value-added method to quantitatively identify the key indicators and then combined these with the fuzzy Delphi method to further ensure the comprehensiveness of the evaluation system. The indicators of the two models were used together to determine the final input–output indicator system.

(1)　　Principle of indicator selection

The value-added method is an important research aspect of value management. It is based on whether all inputs in the balance sheet generate value or destroy value as an input or output. However, industrial research is different from research on enterprises. No balance sheet can be referenced, and input–output indicators are not as clearly defined as they are in the types of costs for enterprises. The input and output index systems constructed by existing construction-related research are often intersected or repeated. Therefore, this study only learns from such analytical ideas and uses factor analysis models to screen the critical indicators. The fuzzy Delphi method combines fuzzy theory and the Delphi method [40,41]. Thus, it evaluates objects or schemes with fuzzy numbers. Combined with expert scoring, the triangle fuzzy numbers are constructed by using the maximum, minimum, and geometric means. The median of each indicator expert's scoring geometric mean is selected as the threshold of the index, and the score is converted into objective data [42]. In this way, the credibility and reasonableness of the evaluation results are ensured.

(2)　　Input and output indicator selection

According to the literature review, Table 1 summarizes the existing input–output variables for the evaluation of the production efficiency of the construction industry. Among them, the input indicators mainly include the number of employees, the number of construction enterprises' technical equipment rate, the power equipment rate, and the total assets of the construction industry. Meanwhile, the output indicators mainly include the construction area, the project completion area, the construction industry's added value, the engineering settlement profit, the total construction output value, and the total profit. Due to the technical and power equipment rate, the interprovincial data were difficult to obtain. In terms of the construction area, the total construction industry output value can be replaced by the area of construction completion and total profit. In addition, the profit of project settlement refers to the profit achieved by the settled projects, the level of which is closely related to the schedule arrangement and resource allocation efficiency, and is a more comprehensive and intuitive reflection of production efficiency. In summary, the study selected six indicators as independent variables: number of enterprises (Enterprise), number of employees (Employees), total assets of the construction industry (Assets), added value of the construction industry (Added), area of completed projects (Completed), and total profit of the construction industry (Profit). This study considered the profit of project settlement as a dependent variable for regression analysis. Based on the absolute value of the regression coefficient size, the key factors that affect the added value of the profit of the construction industry were found. The model also investigated the influence of each variable on the change in the added value of the profit of the construction industry through its direction. As shown in the regression results in Table 2, the value added of the construction industry, the number of construction industry enterprises, and the total assets of the construction industry are the main factors affecting the value added of the profit of construction settlements.

**Table 2.** Results of the regression analysis.

| Variable | Coefficient | Variable | Coefficient |
|---|---|---|---|
| Added | 0.036 ** | Completed | 32.708 |
|  | (0.017) |  | (13.611) |
| Enterprise | 208.545 *** | Profit | 0.215 |
|  | (46.282) |  | (1.582) |
| Employees | −0.025 | Constant | −141,616.858 |
|  | (0.129) |  | (118,006.538) |
| Assets | 0.017 *** | Observations | 248 |
|  | (0.002) | R-squared | 0.900 |

*** $p < 0.01$, ** $p < 0.05$.

In order to ensure the rationality of the constructed post–input–output index system, this study also uses the Delphi method to conduct the secondary screening of input–output indicators to the screening of indicators through the value-added method. The detailed processing steps are as follows: (1) Based on the preliminary indicator system obtained above (number of enterprises, number of employees, total assets of the construction industry, value added of the construction industry, area of completed projects, total profit of the construction industry, and profit of the project settlement), the questionnaire of the input–output indicator structure of the productivity of the construction industry was prepared. (2) The questionnaire for preliminary research was distributed. Then, the questionnaire was revised and improved based on the research results, and the final questionnaire was prepared. This was then distributed to relevant experts in order to obtain professional and authoritative opinions and suggestions. In this study, the research was conducted via email questionnaire, and the 12 experts included 6 master and doctoral students in the field of construction economics and management, 3 engineers working in the construction industry, and 3 professors in the construction economics and management area. A total of 12 valid questionnaires were collected in the study. (3) Based on multiple rounds of research, the index system of the input–output of productivity in the construction industry was determined through statistical analysis. Due to space limitation, the calculation process was omitted. The final Delphi method determined the input indicators as the number of enterprises in the construction industry (Enterprise) and the total assets of the construction industry (Assets) and the output indicators as the value added of the total output value of the construction industry (Added) and the area of completed projects (Completed). The results of the two models were combined, and the number of enterprises and assets in the construction industry was used as input indicators, while the value added of the total construction output value and completed area was used as output indicators. In this paper, the panel data of 31 provinces and autonomous regions from 2010 to 2020 were used as the research sample, and the data were compiled from *China Construction Industry Yearbook* and *China Statistical Yearbook*, and the missing values were completed using the gray prediction model. For the comparative analysis of regional construction industry productivity, the 31 provinces, municipalities, and autonomous regions were divided into eastern, central, and western for analysis. Eastern includes Beijing, Tianjin, Hebei, Shandong, Liaoning, Shanghai, Jiangsu, Zhejiang, Fujian, Guangdong, and Hainan; central includes Henan, Hubei, Hunan, Anhui, Jiangxi, Shanxi, Jilin, and Heilongjiang; and western includes Sichuan, Chongqing, Inner Mongolia, Guangxi, Yunnan, Ningxia, Shaanxi, Guizhou, Gansu, Qinghai, Tibet, and Xinjiang.

3.1.2. Evaluation Model Establishment and Data Analysis

In order to measure the productivity of the construction industry more comprehensively, this study examines the productivity of the construction industry from both static and dynamic measurement approaches. First, we integrated the scale effect, limited resources, and overall comparability parameters. Then, we chose the nonradial, nonoriented, and variable returns to scale (VRS) bootstrap-DEA model. Compared with the traditional static DEA measurement method, this model excludes the interference of the random factors and errors that are caused by omitted variables, and it can more accurately estimate the static evaluation results of the regional construction industry's productivity covering 31 provinces, municipalities, and autonomous regions across China. Additionally, then, the study was based on the Malmquist-DEA model, which is used to calculate the dynamic production efficiency values of the construction industry. This idea can effectively reduce the errors in the evaluation results, which are often from random shocks, such as statistical errors and omitted data variables, and which decompose the construction productivity from a dynamic perspective.

(1) DEA

Data envelopment analysis (DEA), the most typical nonparametric method, is widely used to evaluate the efficiency of different systems. DEA uses mathematical planning

models to evaluate the relative effectiveness (called DEA validity) between "departments" or "units" (called decision-making units, abbreviated as DMUs) with multiple inputs, especially multiple outputs, and to determine whether a DMU is DEA valid based on the data observed for each DMU. This method can effectively reduce the subjectivity of uncertainty in the calculation process [43]. The conventional DEA model has certain deficiencies. On the one hand, when the observed samples are limited, the DEA estimation results are highly susceptible to random factors, and there are obvious sample sensitivities. On the other hand, the conventional DEA model estimates ignore statistical inference and random error problems; moreover, there are often small samples used. The biased problem leads to a certain deviation in the evaluation value of production efficiency. The bootstrap-DEA model proved to be effective in correcting this shortcoming [21]. Later, more research gradually applied the bootstrap model to the DEA model [44,45] and proposed a mathematical variant of the bootstrap method in order to correct the DEA method. Bootstrap techniques based on repeated self-sampling can provide a more accurate measure of the estimated efficiency of traditional DEA models and its variation.

(2)　Bootstrap-DEA correction measure model

The bootstrap-DEA estimation model can eliminate the influence of extreme values, random errors, and missing variables on the efficiency measurement results. Giving a statistical estimate of the efficiency can make the efficiency evaluation and analysis results more accurate. Thus, by considering that, this study designs a bootstrap-DEA model based on a variable-scale, nonradial, and nonoriented approach in order to measure the static production efficiency of a regional construction industry in China. In the evaluation result, if the efficiency value is less than 1, the decision unit does not reach the optimal production efficiency; if the efficiency value is equal to 1, it indicates that the evaluated decision unit is strong and effective. The model implementation steps are as follows:

Step 1: Calculate the original efficiency of each decision unit $DMU_k$ ($k$ = 1, 2, 3, . . . , $n$) using the traditional DEA measurement model. Extract a simple sample of size $n$ using the repeated sampling bootstrap method. Further, $b$ represents the number of iterations of the bootstrap sample. The sample collection is as follows:

$$\hat{\theta}^b = \left\{ \hat{\theta}^b_k | k = 1, 2, \ldots, n \right\} \tag{1}$$

Step 2: Use the kernel density estimation method to smooth the sample that is obtained by the plain Bootstrap method. Then, obtain the sample according to the smoothing bootstrap method to correct the original sample input index. This adjusted calculation is as follows:

$$x^b_k = \left( \frac{\hat{\theta}_k}{\overline{\theta}_k^b} \right) x_k (k = 1, \ldots, n) \tag{2}$$

According to the bootstrap-adjusted input data and the initial output data as a new sample, the traditional DEA method is used to recalculate the efficiency value:

$$\widetilde{\theta}^b = \left\{ \widetilde{\theta}_k^{\,b} | k = 1, \ldots, n \right\} \tag{3}$$

Step 3: Repeat steps 1 and 2 to obtain a series of efficiency values, and then calculate the correction efficiency value deviation and correct DEA efficiency value of each decision unit $DMU_k$ ($k$ = 1, 2, 3, . . . , $n$). The expression is as follows:

$$Bias_k = E\left( \widetilde{\theta}_k^{\,b} \right) - \widetilde{\theta}_k = \frac{1}{B} \sum_{b=1}^{B} \widetilde{\theta}_k^{\,b} - \hat{\theta}_k \tag{4}$$

$$\widetilde{\theta}_k = \overset{\wedge}{\theta}_k - Bias = 2\overset{\wedge}{\theta}_k - \frac{1}{B}\sum_{b=1}^{B}\widetilde{\theta}_k^{\,b} \tag{5}$$

In the above formula, $Bias_k$ is the deviation of the correction efficiency value.

(3) Malmquist exponential decomposition method

The DEA-Malmquist index method effectively solves this problem and, as an extension of the DEA model, it can be used not only to measure the change in total factor productivity of DMUs over time but also to avoid the assumptions made in the calculation of Solow residuals. It can also avoid the assumptions made in the calculation of Solow residuals and can be decomposed into the product of efficiency improvements and technological progress, resulting in a more scientific dynamic analysis. Based on the variable assumption of factor size return, Banker et al. (2004) proposed a BCC model to further decompose technical efficiency into pure technical efficiency and scale efficiency [46]. This method has been widely used [47–49]. In the DEA-Malmquist index method, the change in TFP between two adjacent data points is measured by estimating the ratio of the distance of each data point to a common boundary of the production possibilities. The measured index is called the Malmquist-TFP index [50], and the output-oriented Malmquist-TFP index of the *i*-th DMU from base period *s* to period *t* can be defined:

$$m_0\left(x_0^s, y_0^s, x_0^t, y_0^t\right) = \left[\frac{d_0^s\left(x_0^t, y_0^t|C\right)}{d_0^s\left(x_0^s, y_0^s|C\right)} \times \frac{d_0^t\left(x_0^t, y_0^t|C\right)}{d_0^t\left(x_0^s, y_0^s|C\right)}\right]^{\frac{1}{2}} \tag{6}$$

In Equation (6), $d_0^s\left(x_0^t, y_0^t|C\right)$ and $d_0^s\left(x_0^s, y_0^s|C\right)$ are the distance functions at constant payoffs to scale. $d_0^s\left(x_0^t, y_0^t|C\right)$ denotes the production point distance function for the *i*-th DMU in period *t*, using the technology in period *t* as a reference, and $d_0^s\left(x_0^s, y_0^s|C\right)$ denotes the production point distance function in period *s*. When $m_0\left(x_0^s, y_0^s, x_0^t, y_0^t\right) > 1$, this indicates that the TFP has a growth from base period *s* to period *t*; when $m_0\left(x_0^s, y_0^s, x_0^t, y_0^t\right) < 1$, this indicates that the TFP has a negative growth from base period *s* to period *t*.

The Malmquist index obtained from the above formula can be decomposed into two categories: the efficiency change (EFF) and the technical change index (Techch), which are expressed as follows:

$$m_0\left(x_0^s, y_0^s, x_0^t, y_0^t\right) = \frac{d_0^s\left(x_0^t, y_0^t|C\right)}{d_0^t\left(x_0^s, y_0^s|C\right)}\left[\frac{d_0^t\left(x_0^s, y_0^s|C\right)}{d_0^s\left(x_0^s, y_0^s|C\right)} \times \frac{d_0^t\left(x_0^t, y_0^t|C\right)}{d_0^s\left(x_0^t, y_0^t|C\right)}\right]^{\frac{1}{2}}$$
$$= EFFch \times TCHch \tag{7}$$

In the case of variable scale compensation, the technical efficiency change index (*EFFch*) can be further broken down into the pure technical efficiency change (*PTEch*) and the scale efficiency change (*SEch*), which can be expressed as follows:

$$Ech = \frac{d_0^t\left(x_0^s, y_0^s|C\right)}{d_0^s\left(x_0^s, y_0^s|C\right)} = \frac{\frac{d_0^t\left(x_0^t, y_0^t|C\right)}{d_0^t\left(x_0^t, y_0^t|V\right)}}{\frac{d_0^s\left(x_0^s, y_0^s|C\right)}{d_0^s\left(x_0^t, y_0^t|V\right)}} \times \frac{d_0^t\left(x_0^t, y_0^t|V\right)}{d_0^s\left(x_0^s, y_0^s|V\right)}$$
$$= \frac{SE_0^t\left(x_0^t, y_0^t\right)}{SE_0^s\left(x_0^s, y_0^s\right)} \times \frac{d_0^t\left(x_0^t, y_0^t|V\right)}{d_0^s\left(x_0^s, y_0^s|V\right)} = SEch \times PTEch \tag{8}$$

$$\begin{aligned} m_0\left(x_0^s, y_0^s, x_0^t, y_0^t\right) &= TFPch = TECHch \times Ech \\ &= TECHch \times SEch \times PTEch \end{aligned} \tag{9}$$

Therefore, the TFP index can be expressed by the product of the technological progress index, the scale efficiency index, and the pure technical efficiency index. Among them, the changes in research and development and the introduction of new technology are reflected by technological advances. The changes in the optimization, promotion, and application of current technologies, and the rational allocation of production factors, are reflected by

pure technical efficiency, management methods, and production scale. Changes in the aforementioned are reflected by scale efficiency. TFP is a comprehensive reflection of the overall changes in these aspects [51].

### 3.1.3. Comprehensive Productivity Calculation and Visualization

According to the previous process, the static production efficiency and the corrected dynamic production efficiency value can be obtained. Then, the two indicators are combined, and different weights are assigned based on the existing research to calculate the comprehensive production efficiency value of the construction industry. Finally, the comprehensive production efficiency results are visualized with the help of the eChart software, and the comprehensive production efficiency results of the construction industry in different regions are compared and analyzed.

### *3.2. Case Study*

This study uses the panel data of 31 provinces and autonomous regions from 2010 to 2020 as the research samples. The data are compiled from the *China Construction Industry Yearbook* and the *China Statistical Yearbook*. The missing values are complemented by the gray prediction model. In order to compare and analyze the production efficiency of the regional construction industry, 31 provinces, municipalities, and autonomous regions were divided into eastern, central, and western regions for the purposes of analysis [9]. These regions represent the different economic development levels. The western region includes 12 provinces: Chongqing, Sichuan, Guizhou, Yunnan, Tibet, Shaanxi, Gansu, Qinghai, Ningxia, Xinjiang, Guangxi, and Inner Mongolia. The midland region consists of Shanxi, Anhui, Jiangxi, Henan, Hubei, Heilongjiang, Jilin, and Hunan. The eastern region refers to the following 10 provinces: Beijing, Tianjin, Hebei, Shanghai, Jiangsu, Zhejiang, Fujian, Shandong, Liaoning, Guangdong, and Hainan, which are located on the eastern coast of China.

## 4. Empirical Results

### *4.1. Static Production Efficiency Comparison Analysis*

In this study, with the help of the MaxDEA software, the bootstrap-based stochastic DEA method is used to correct the production efficiency of China's construction industry to obtain the "true value" of the production efficiency. According to the ideology of the bootstrap-DEA model, the DEA estimator is measured by repeated sampling and experience distribution. In general, with the increase in the number of bootstrap iterations, the accuracy of the calculation results will be more accurate. The number of iterations is 1000, 4000, and 5000, and the confidence level is 0.012, 0.015, and 0.001. The results are not much different. The average efficiency value before and after the correction and the related output results are shown in Table 3.

The comparison results show that although the efficiency rankings before and after correction have not changed much, the average efficiency values after correction by each decision unit are lower than the average efficiency values measured by the traditional DEA model; further, all the average errors are greater than zero. Table 3 shows the confidence intervals for the regional construction industry productivity based on the bootstrap-DEA method. The average efficiency value calculated by the traditional DEA model was found to be outside the confidence interval, and the calculated results after correction were within the confidence interval. In general, the confidence interval can effectively reflect the actual efficiency value. If the estimation result falls outside the confidence interval, then the estimation result of the traditional DEA model can be biased [52]. The traditional DEA method is highly dependent on the original data and cannot show the characteristics of nonparametric statistics. The results of the bootstrap-DEA model are more reliable and real. This study uses the corrected production efficiency value of the construction industry for subsequent analysis.

**Table 3.** Comparison results of the two models.

| DMU | Average Efficiency | Precorrection Ranking | Corrected Average Efficiency | Revised Ranking | Average Bias | Confidence Interval Lower Limit | Confidence Interval Upper Limit |
|---|---|---|---|---|---|---|---|
| Beijing | 0.2359 | 28 | 0.2190 | 28 | 0.0169 | 0.1748 | 0.2954 |
| Tianjin | 0.2268 | 29 | 0.2079 | 29 | 0.0189 | 0.1546 | 0.2099 |
| Hebei | 0.4827 | 15 | 0.4543 | 14 | 0.0283 | 0.2386 | 0.3817 |
| Shanxi | 0.2032 | 31 | 0.1856 | 31 | 0.0176 | 0.1257 | 0.1668 |
| Inner Mongolia | 0.4629 | 17 | 0.4427 | 15 | 0.0203 | 0.2026 | 0.2815 |
| Liaoning | 0.3724 | 23 | 0.3296 | 23 | 0.0428 | 0.1186 | 0.1622 |
| Jilin | 0.4296 | 19 | 0.4228 | 17 | 0.0068 | 0.1959 | 0.2688 |
| Heilongjiang | 0.3290 | 26 | 0.3228 | 25 | 0.0061 | 0.1572 | 0.2123 |
| Shanghai | 0.2464 | 27 | 0.2268 | 27 | 0.0196 | 0.1872 | 0.2844 |
| Jiang Su | 1.0000 | 1 | 0.6741 | 6 | 0.3259 | 0.3992 | 0.9531 |
| Zhejiang | 1.0000 | 1 | 0.6477 | 8 | 0.3523 | 0.4126 | 0.9425 |
| Anhui | 0.5932 | 10 | 0.5403 | 9 | 0.0528 | 0.2447 | 0.5007 |
| Fujian | 1.0000 | 1 | 0.9365 | 1 | 0.0598 | 0.6947 | 1.3249 |
| Jiangxi | 0.8204 | 6 | 0.7625 | 4 | 0.0401 | 0.3238 | 0.6434 |
| Shandong | 0.4476 | 18 | 0.3534 | 22 | 0.0941 | 0.1475 | 0.3765 |
| Henan | 0.5195 | 14 | 0.4410 | 16 | 0.0786 | 0.2070 | 0.4734 |
| Hubei | 0.6174 | 9 | 0.4885 | 13 | 0.1288 | 0.2936 | 0.6240 |
| Hunan | 0.7891 | 7 | 0.6949 | 5 | 0.0941 | 0.4516 | 0.7580 |
| Guangdong | 0.3602 | 24 | 0.2955 | 26 | 0.0646 | 0.1197 | 0.3420 |
| Guangxi | 0.6828 | 8 | 0.6561 | 7 | 0.0266 | 0.4665 | 0.7671 |
| Hainan | 1.0000 | 1 | 0.8005 | 2 | 0.1995 | 0.5478 | 2.0721 |
| Chongqing | 0.5715 | 11 | 0.5127 | 12 | 0.0587 | 0.3093 | 0.5700 |
| Sichuan | 0.4767 | 16 | 0.4184 | 18 | 0.0583 | 0.1656 | 0.4173 |
| Guizhou | 0.3979 | 22 | 0.3854 | 21 | 0.0124 | 0.3051 | 0.4324 |
| Yunnan | 0.3400 | 25 | 0.3273 | 24 | 0.0127 | 0.2074 | 0.3047 |
| Tibet | 0.9143 | 5 | 0.7720 | 3 | 0.0950 | 0.0723 | 0.5560 |
| Shaanxi | 0.5619 | 12 | 0.5361 | 10 | 0.0462 | 0.2129 | 0.3388 |
| Gansu | 0.4292 | 20 | 0.4184 | 18 | 0.0108 | 0.2284 | 0.3059 |
| Qinghai | 0.2191 | 30 | 0.1919 | 30 | 0.0272 | 0.1100 | 0.2266 |
| Ningxia | 0.4016 | 21 | 0.3980 | 20 | 0.0037 | 0.1583 | 0.2520 |
| Xinjiang | 0.5466 | 13 | 0.5267 | 11 | 0.0200 | 0.2964 | 0.4034 |

In order to more intuitively display the difference between the results obtained before and after the correction, Figure 2 compares the average production efficiency values calculated by the two models. In addition, it shows that the variation trend of the production efficiency value curves that are calculated by the two methods is the same. However, great differences appear at the peak, and the production efficiency value after the correction by the bootstrap-DEA model is significantly smaller than that obtained by the traditional DEA model. It can be seen from the combination in Figure 2 that 4 of the 31 decision-making units are valid for DEA, namely, Jiangsu, Zhejiang, Fujian, and Hainan, whereas the remaining 27 provinces and cities are invalid. The proportion of DEA effective provinces and cities is only 13%. In addition, the average efficiency value of the construction industry in the central and western regions is significantly larger than that in the eastern region. The efficiency values before and after correction in Beijing, Tianjin, Shanghai, and Guangdong are all below 0.5.

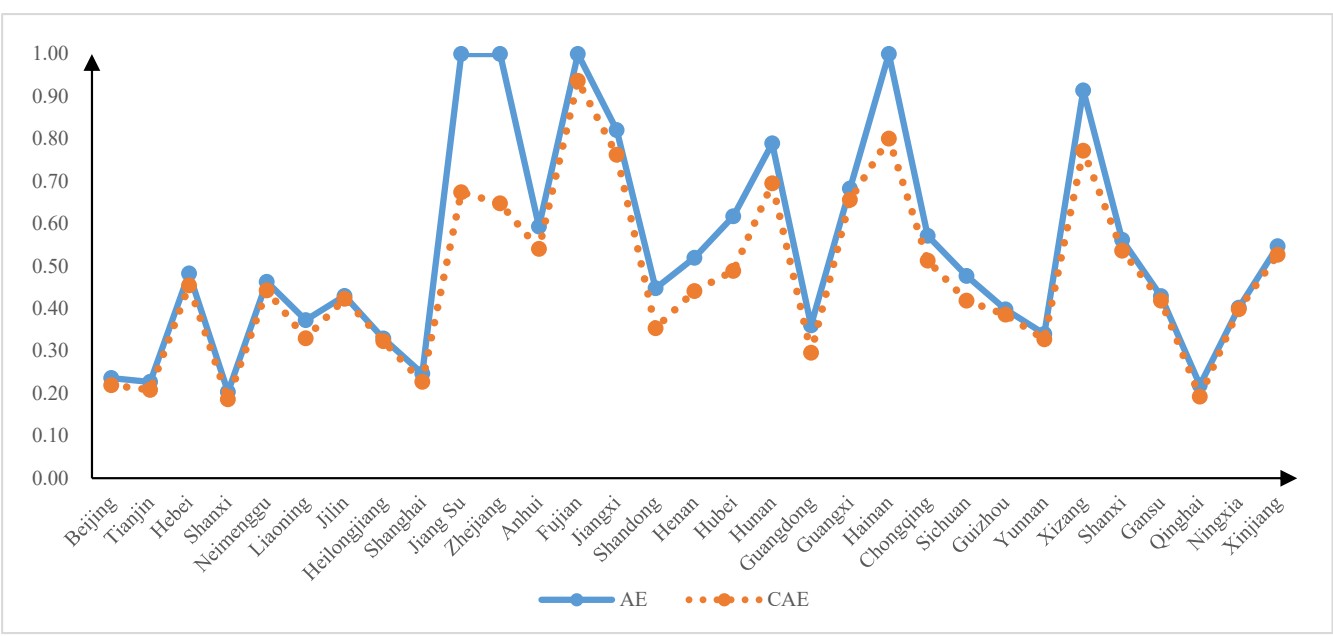

**Figure 2.** Comparative analysis of average production efficiency in 2010–2020.

*4.2. Static Production Efficiency Measurement*

As can be seen from Table 4 and Figure 3, Hainan, Fujian, Jiangxi, Hunan, Guangxi, Zhejiang, Jiangsu, Tibet, Shaanxi, Anhui, and other places are the top 10 regarding production efficiency of the construction industry. Hainan, Jiangxi, Hunan, and Guangxi are the less developed regions in the central and western regions. Additionally, the construction industry there started late, and its scale was small at first. However, it has now attached great importance to the efficiency of input and output, and it has taken a route that is in line with its own characteristics, such that the production efficiency is satisfying. In the developed eastern regions, such as Beijing, Shanghai, Tianjin, and Guangdong, production efficiency is moving backwards. Additionally, according to the regional division, the leading static production efficiency level of China's construction industry is the central region with 0.4693, followed by the western region with 0.4592 and the eastern region (with the lowest static production efficiency value) with 0.4404. The study further confirms the prominent contradiction between the expansion of the construction industry and the improvement of the quality of development. The reason for this phenomenon may be the existence of the redundancy of production factors. At present, the development of China's construction industry mainly relies on factor inputs, external demand, and scale expansion. It has not yet fundamentally moved away from the quantitative growth model [18,19], and the inefficient production management style does not match the scale of the construction industry, thereby leading to the redundancy of production factors and uneconomic scale.

In this study, the production efficiency values of the construction industry in each province from 2010 to 2020 are visualized to form a heat map of production efficiency values. The production efficiency values are of provinces, such as Fujian, Zhejiang, Guangxi, and Jiangsu. In the last decade, they have been stable and have maintained a high level, with the average value greater than 0.6. Meanwhile, Beijing, Shanghai, Guangdong, and other provinces' production efficiency value in the last decade has been maintained at a low level, with the average value less than 0.3. In addition, it is worth noting that Tibet, Shanxi, Fujian, Jiangxi, Xinjiang, and other places' production efficiency values in recent years have shown a declining trend.

**Table 4.** Production efficiency values of the construction industry after bootstrap-DEA correction in 31 provinces from 2010 to 2020.

| DMU | 2010 | 2011 | 2012 | ... | 2018 | 2019 | 2020 | AE | Rank |
|---|---|---|---|---|---|---|---|---|---|
| Beijing | 0.1749 | 0.1823 | 0.2119 | ... | 0.2414 | 0.0356 | 0.1046 | 0.1940 | 30 |
| Tianjin | 0.2127 | 0.2139 | 0.1987 | ... | 0.3542 | 0.4171 | 0.4515 | 0.2624 | 27 |
| Hebei | 0.5308 | 0.5688 | 0.4996 | ... | 0.2846 | 0.4455 | 0.4832 | 0.4407 | 14 |
| Shanxi | 0.2249 | 0.1910 | 0.1980 | ... | 0.1203 | 0.1211 | 0.2396 | 0.1787 | 31 |
| Inner Mongolia | 0.6596 | 0.6158 | 0.4674 | ... | 0.2246 | 0.3120 | 0.4083 | 0.4078 | 17 |
| Liaoning | 0.4647 | 0.4763 | 0.3980 | ... | 0.1048 | 0.1718 | 0.2785 | 0.2902 | 26 |
| Jilin | 0.6666 | 0.5908 | 0.4552 | ... | 0.2600 | 0.3257 | 0.3893 | 0.3962 | 19 |
| Heilongjiang | 0.5316 | 0.4024 | 0.3495 | ... | 0.1906 | 0.2210 | 0.2481 | 0.2948 | 25 |
| Shanghai | 0.2396 | 0.2238 | 0.2161 | ... | 0.2358 | 0.2009 | 0.1499 | 0.2183 | 29 |
| Jiangsu | 0.7283 | 0.7286 | 0.6904 | ... | 0.5494 | 0.4729 | 0.3786 | 0.6176 | 7 |
| Zhejiang | 0.6702 | 0.6754 | 0.6417 | ... | 0.6063 | 0.6035 | 0.5929 | 0.6350 | 6 |
| Anhui | 0.6126 | 0.6203 | 0.5556 | ... | 0.3954 | 0.3703 | 0.4761 | 0.5059 | 10 |
| Fujian | 1.000 | 0.9526 | 0.9146 | ... | 0.8064 | 0.6360 | 0.5470 | 0.8620 | 2 |
| Jiangxi | 0.6601 | 1.000 | 1.000 | ... | 0.5484 | 0.6831 | 0.7783 | 0.7372 | 3 |
| Shandong | 0.3956 | 0.3861 | 0.3628 | ... | 0.3181 | 0.3639 | 0.5050 | 0.3650 | 21 |
| Henan | 0.5114 | 0.5019 | 0.4552 | ... | 0.5586 | 0.6121 | 0.6952 | 0.4903 | 12 |
| Hubei | 0.4721 | 0.5206 | 0.4863 | ... | 0.4394 | 0.4523 | 0.4632 | 0.4785 | 13 |
| Hunan | 0.7132 | 0.7058 | 0.6542 | ... | 0.6053 | 0.6101 | 0.6292 | 0.6731 | 4 |
| Guangdong | 0.3004 | 0.3275 | 0.3184 | ... | 0.2751 | 0.2968 | 0.3300 | 0.2969 | 24 |
| Guangxi | 0.6508 | 0.6521 | 0.5781 | ... | 0.6616 | 0.6714 | 0.6864 | 0.6607 | 5 |
| Hainan | 0.7022 | 0.7447 | 0.7027 | ... | 0.9970 | 1.0447 | 1.1144 | 0.8691 | 1 |
| Chongqing | 0.5530 | 0.5059 | 0.4936 | ... | 0.4597 | 0.4382 | 0.4213 | 0.4928 | 11 |
| Sichuan | 0.4565 | 0.4348 | 0.4416 | ... | 0.3202 | 0.3189 | 0.4499 | 0.4033 | 18 |
| Guizhou | 0.3743 | 0.4104 | 0.3743 | ... | 0.3195 | 0.1846 | 0.1146 | 0.3366 | 23 |
| Yunnan | 0.3594 | 0.3371 | 0.3509 | ... | 0.3168 | 0.3917 | 0.4862 | 0.3467 | 22 |
| Tibet | 1.000 | 0.8557 | 0.8324 | ... | 0.2204 | 0.0829 | 0.1170 | 0.5997 | 8 |
| Shaanxi | 1.000 | 1.0000 | 0.4348 | ... | 0.4379 | 0.5795 | 0.6836 | 0.5445 | 9 |
| Gansu | 0.5023 | 0.5485 | 0.4146 | ... | 0.3477 | 0.4729 | 0.5592 | 0.4297 | 16 |
| Qinghai | 0.1989 | 0.2265 | 0.1833 | ... | 0.2889 | 0.3939 | 0.4265 | 0.2404 | 28 |
| Ningxia | 0.4806 | 0.5906 | 0.4673 | ... | 0.2482 | 0.3776 | 0.4863 | 0.3905 | 20 |
| Xinjiang | 0.5337 | 0.5673 | 0.5656 | ... | 0.2305 | 0.1830 | 0.1238 | 0.4319 | 15 |
| Eastern | 0.4927 | 0.4982 | 0.4686 | ... | 0.4339 | 0.4262 | 0.4487 | 0.4592 | 2 |
| Central | 0.5491 | 0.5666 | 0.5192 | ... | 0.3898 | 0.4245 | 0.4899 | 0.4693 | 1 |
| Western | 0.5641 | 0.5621 | 0.4670 | ... | 0.3397 | 0.3672 | 0.4136 | 0.4404 | 3 |

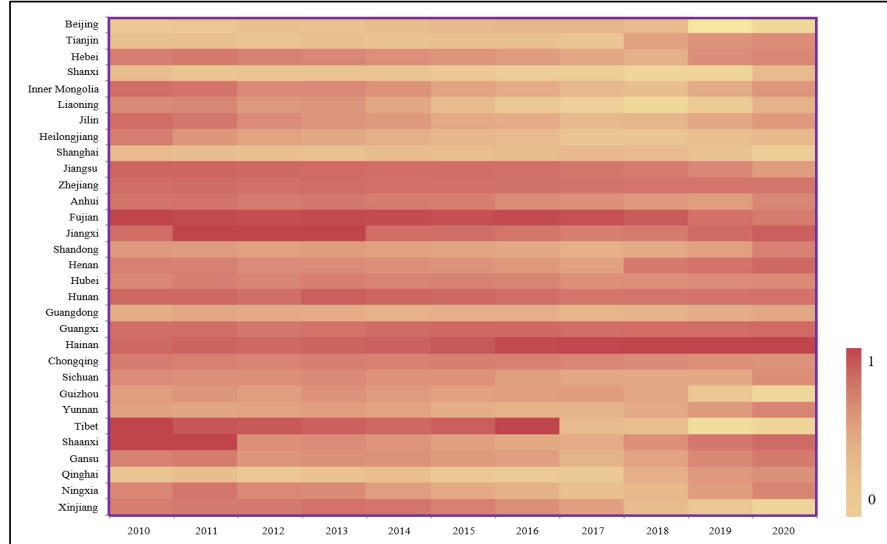

**Figure 3.** Heat map of the static production efficiencies of the construction industry in 31 provinces in China.

### 4.3. Dynamic Production Efficiency Measures

In order to improve the evaluation effect and to consider the change in the growth rate of the production factor input and technological progress in the construction industry, this study combines the dynamic perspective DEA-Malmquist model to reveal the change in the decomposition index of the production efficiency variation of the construction industry. The calculation results are shown in Table 5 and Figure 4.

**Table 5.** The decomposition index of the DEA-Malmquist model in 2010–2020.

| Year | Technical Efficiency Change | Technological Progress Index | Pure Technical Efficiency Change | Scale Efficiency Index | MI Index |
|---|---|---|---|---|---|
| 2010–2011 | 1.0450 | 1.0570 | 1.0490 | 0.9960 | 1.1050 |
| 2011–2012 | 0.9640 | 1.1420 | 0.9630 | 1.0010 | 1.1000 |
| 2012–2013 | 1.0620 | 1.0690 | 1.0860 | 0.9780 | 1.1350 |
| 2013–2014 | 0.9330 | 1.0760 | 0.9580 | 0.9740 | 1.0040 |
| 2014–2015 | 0.9950 | 1.0030 | 0.9910 | 1.0040 | 0.9980 |
| 2015–2016 | 0.9880 | 1.0110 | 0.9880 | 1.0000 | 0.9990 |
| 2016–2017 | 0.9400 | 1.0180 | 0.9330 | 1.0070 | 0.9570 |
| 2017–2018 | 0.9320 | 1.0093 | 0.9251 | 0.9984 | 0.9489 |
| 2018–2019 | 0.9427 | 0.9647 | 0.9427 | 0.9542 | 0.9532 |
| 2019–2020 | 0.9305 | 0.9380 | 0.9267 | 0.9389 | 0.9333 |
| Average value | 0.9732 | 1.0288 | 0.9763 | 0.9852 | 1.0133 |

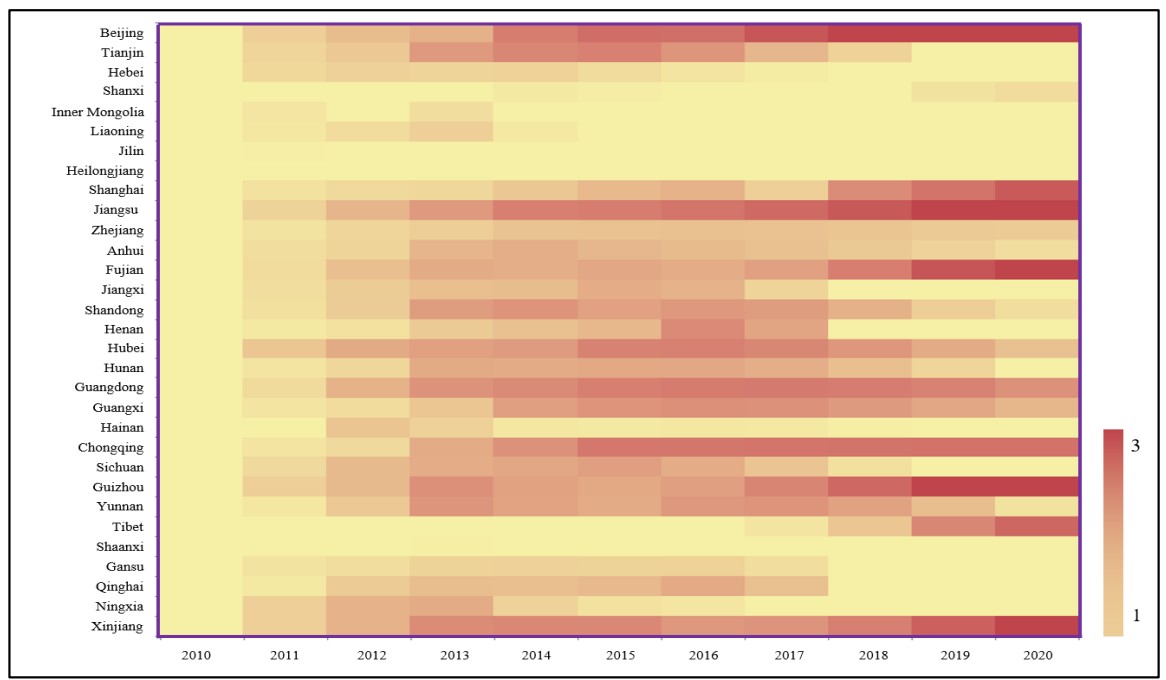

**Figure 4.** Heat map of the dynamic production efficiencies of the construction industry in 31 provinces in China.

The results show that, overall, the average value of the total factor productivity index for China's construction industry was 1.0133, indicating an average annual increase in total factor productivity of 1.33%. The national average technical efficiency index was 0.9732, the technical progress index was 1.0288, and the scale efficiency index was 0.9852. The gap between the technical efficiency level of each decision unit and the frontier area widened, and the decline in the average technical efficiency was mainly determined by the pure technical efficiency. Among them, the average pure technical efficiency index was 0.9763, indicating that the optimal allocation of technical inputs and outputs had not yet been reached, and the efficiency of scientific and technological inputs had not yet been

coordinated with the development level of the construction industry, which is consistent with the findings of other scholars. The relative growth rate of technological progress was 2.88%. This indicates that technological progress is the main driver of productivity improvement in China's construction industry. In terms of annual slices, the scale efficiency index of China's construction industry showed a fluctuating upward trend from 2010 to 2013, indicating to some extent that the scale efficiency of China's construction industry has gradually improved. However, it is worth noting that the technical efficiency and scale efficiency indicators, which measure the resource allocation capacity and resource utilization efficiency of the construction industry during this period, show an opposite trend to scale efficiency. This indicates that as the scale efficiency of the construction industry increases, the construction industry has not achieved intensive and efficient development at this time. In 2013–2020, the scale efficiency, technical efficiency, and pure technical efficiency of the construction industry all showed a fluctuating downward trend, with the lowest level in 2019–2020. This indicates that China's construction industry made good use of the production factors invested in the early stage, and later, due to the impact of the reform of the institutional mechanism of China's construction industry, some production factors were idle and used inefficiently. The increase in scale efficiency in the construction industry did not lead to an increase in management efficiency and the accumulation of production experience. Then, the average scale efficiency index during the study period is 0.9852, with a scale efficiency less than 1. The scale efficiency index showed a fluctuating downward trend during the study period. The highest value was in 2016–2017, when the scale efficiency index was 1.007, and the lowest value was in 2019–2020, when the scale efficiency index was 0.9389. Overall, the factor input of China's construction industry is in a state of diseconomies of scale, and the driving effect of purely increasing output on the productivity of the construction industry is gradually weakening, relying on technological innovation to promote the improvement of the total factor productivity level of the construction industry. In 2017–2020, the scale efficiency decreased year by year, indicating that the factor inputs of the construction industry are gradually saturated. The main reasons for this may be the competition and fragmentation among different regions, the local excess of inputs in the construction industry leads to overall diminishing returns to scale, and the scale economies embedded in the input factors of production are not fully exploited.

To create a heat map of productivity levels, this study visualized the productivity levels of the construction industry for each province from 2010 to 2020. Over the past decade, Inner Mongolia, Liaoning, Jilin, Heilongjiang, and Shaanxi have been at a low level, with dynamic efficiency values of less than 1, well below the average dynamic efficiency value of 1.3572. The three cities with the highest average dynamic efficiency values are Beijing, Jiangsu, and Guizhou, with dynamic efficiency values greater than 2. In addition, it is worth noting that in terms of the development of dynamic productivity values in construction, Beijing, Shanghai, Jiangsu, Zhejiang, Fujian, Hubei, Chongqing, Guizhou, Tibet, and Xinjiang have the highest dynamic efficiency values. In addition, it is worth noting that in terms of the development trend of the dynamic productivity value of construction, Beijing, Shanghai, Jiangsu, Zhejiang, Fujian, Hubei, Chongqing, Guizhou, Tibet, and Xinjiang show a gradual increase in the dynamic productivity value of construction, and these cities are mainly eastern and central cities. On the other hand, Hebei, Shandong, Sichuan, Qinghai, Ningxia, and Yunnan, which are mainly cities in the western region, show a gradual decline in construction productivity. It can be seen that the total factor productivity growth shows obvious fluctuation (stage) characteristics and unbalanced growth among regions, mainly showing that the total factor productivity growth of the construction industry in the eastern region is higher than that of the central and western regions and the national average, and the total factor productivity growth of China's construction industry basically shows a decreasing trend in the order from the eastern to the central region and then to the west.

### 4.4. Comprehensive Production Efficiency Measurement

According to the static and dynamic production efficiency values of the construction industry in each province, i.e., the city and autonomous regions obtained above, it can be found that there are differences in the efficiency ranking of each decision-making unit. In order to evaluate the comprehensive level and change of production efficiency in the construction industries in various provinces, municipalities, and autonomous regions, this study combines the static and dynamic efficiency values to estimate the comprehensive value of the construction industry's production efficiency more reasonably. The model design is as follows:

$$\text{CPE} = \text{Bootstrap} - \text{DEA} \times \alpha + \text{MI index} \times \beta \tag{10}$$

The bootstrap-DEA method mainly evaluates the efficiency of resource allocation in the construction industry. The Malmquist index mainly evaluates the changes in production efficiency. However, due to the different development levels of the base period, the growth rates of the input and output of the various factors vary greatly among the provinces, which may cause distortion in the efficiency evaluation results. According to the research objectives of this study, based on the theory of objective and fuzzy decision making, the bootstrap-DEA static preference coefficient $\alpha$ is set to 0.8, and the MI index dynamic preference coefficient $\beta$ is set to 0.2 [7]. This is performed in order to calculate these values in the 31 provinces and municipalities in China from 2010 to 2020. China's construction industry's comprehensive production efficiency value and its comprehensive estimated efficiency value are thus sorted. The results are shown in Table 6 and Figure 5. Except for the comprehensive production efficiency of certain years in Fujian, Jiangxi, Tibet, Hainan, and Shaanxi, which is greater than or equal to 1, the overall production efficiency of the construction industry in all provinces in 2010–2020 is less than 1, and the resource allocation is not effective. Additionally, there are significant differences between the decision-making units. Differences should strengthen the flow of regional factors and coordinated development. In addition, different from the static and dynamic production efficiency evaluation results above, the comprehensive production efficiency evaluation value shows that the subregional production efficiency level is leading in the east, second in the west, and the middle in the center. The estimated results do neutralize the static and dynamic efficiency values, thus making the evaluation more reasonable.

**Table 6.** CPE of the construction industry in 31 provinces.

| DMU | 2010 | 2011 | 2012 | ... | 2018 | 2019 | 2020 | AE | Rank |
|---|---|---|---|---|---|---|---|---|---|
| Beijing | 0.3399 | 0.4036 | 0.4621 | ... | 0.8244 | 0.9048 | 1.0932 | 0.6522 | 14 |
| Tianjin | 0.3702 | 0.4193 | 0.4302 | ... | 0.5367 | 0.5278 | 0.5192 | 0.4956 | 24 |
| Hebei | 0.6246 | 0.6979 | 0.6554 | ... | 0.4224 | 0.5440 | 0.5618 | 0.5725 | 21 |
| Shanxi | 0.3799 | 0.3410 | 0.3554 | ... | 0.2971 | 0.3210 | 0.4279 | 0.3474 | 31 |
| Inner Mongolia | 0.7277 | 0.7084 | 0.5739 | ... | 0.3110 | 0.3624 | 0.4124 | 0.4875 | 26 |
| Liaoning | 0.5718 | 0.5961 | 0.5540 | ... | 0.1759 | 0.2237 | 0.2844 | 0.3946 | 29 |
| Jilin | 0.7333 | 0.6736 | 0.5252 | ... | 0.2903 | 0.3274 | 0.3693 | 0.4412 | 27 |
| Heilongjiang | 0.6253 | 0.4587 | 0.4294 | ... | 0.2493 | 0.2772 | 0.3081 | 0.3674 | 30 |
| Shanghai | 0.3917 | 0.4057 | 0.4129 | ... | 0.5698 | 0.6065 | 0.6456 | 0.4916 | 25 |
| Jiangsu | 0.7827 | 0.8341 | 0.8590 | ... | 0.9726 | 0.9826 | 0.9926 | 0.9201 | 2 |
| Zhejiang | 0.7361 | 0.7618 | 0.7605 | ... | 0.7622 | 0.7512 | 0.7404 | 0.7689 | 7 |
| Anhui | 0.6901 | 0.7286 | 0.6943 | ... | 0.5865 | 0.5488 | 0.6135 | 0.6726 | 12 |
| Fujian | 1.0000 | 0.9969 | 1.0188 | ... | 1.0571 | 1.0547 | 1.0523 | 1.0473 | 1 |
| Jiangxi | 0.7281 | 1.0320 | 1.0649 | ... | 0.6042 | 0.6375 | 0.6782 | 0.8145 | 5 |
| Shandong | 0.5165 | 0.5359 | 0.5547 | ... | 0.5690 | 0.5525 | 0.6364 | 0.5893 | 18 |
| Henan | 0.6091 | 0.6119 | 0.5903 | ... | 0.6441 | 0.6404 | 0.6867 | 0.6366 | 15 |
| Hubei | 0.5777 | 0.6911 | 0.7156 | ... | 0.7146 | 0.6855 | 0.6576 | 0.7165 | 10 |

**Table 6.** *Cont.*

| DMU | 2010 | 2011 | 2012 | ... | 2018 | 2019 | 2020 | AE | Rank |
|---|---|---|---|---|---|---|---|---|---|
| Hunan | 0.7706 | 0.7828 | 0.7699 | ... | 0.7731 | 0.7360 | 0.7007 | 0.8139 | 6 |
| Guangdong | 0.4403 | 0.5002 | 0.5665 | ... | 0.6378 | 0.6369 | 0.6359 | 0.5960 | 17 |
| Guangxi | 0.7206 | 0.7418 | 0.6983 | ... | 0.8861 | 0.8685 | 0.8513 | 0.8362 | 4 |
| Hainan | 0.7618 | 0.7929 | 0.8389 | ... | 0.9997 | 1.0212 | 1.0663 | 0.9083 | 3 |
| Chongqing | 0.6424 | 0.6241 | 0.6345 | ... | 0.8142 | 0.8014 | 0.7888 | 0.7591 | 8 |
| Sichuan | 0.5652 | 0.5876 | 0.6406 | ... | 0.4837 | 0.4300 | 0.4922 | 0.5829 | 19 |
| Guizhou | 0.4995 | 0.5879 | 0.5961 | ... | 0.7379 | 0.7838 | 0.8325 | 0.6693 | 13 |
| Yunnan | 0.4875 | 0.4837 | 0.5529 | ... | 0.5948 | 0.6039 | 0.6132 | 0.5772 | 20 |
| Tibet | 1.0000 | 0.8428 | 0.8101 | ... | 0.4507 | 0.4566 | 0.5822 | 0.7083 | 11 |
| Shaanxi | 1.0000 | 0.9954 | 0.4989 | ... | 0.5121 | 0.6312 | 0.7228 | 0.6099 | 16 |
| Gansu | 0.6018 | 0.6602 | 0.5635 | ... | 0.4671 | 0.5182 | 0.5749 | 0.5576 | 22 |
| Qinghai | 0.3591 | 0.3930 | 0.4133 | ... | 0.4291 | 0.4243 | 0.4196 | 0.4246 | 28 |
| Ningxia | 0.5845 | 0.7209 | 0.6329 | ... | 0.3027 | 0.3547 | 0.4158 | 0.5012 | 23 |
| Xinjiang | 0.6269 | 0.7120 | 0.7636 | ... | 0.5951 | 0.6544 | 0.7197 | 0.7269 | 9 |
| Eastern | 0.5941 | 0.6313 | 0.6466 | ... | 0.6843 | 0.7096 | 0.7480 | 0.6760 | 1 |
| Central | 0.6393 | 0.6650 | 0.6431 | ... | 0.5199 | 0.5217 | 0.5552 | 0.6013 | 3 |
| Western | 0.6513 | 0.6715 | 0.6149 | ... | 0.5487 | 0.5741 | 0.6188 | 0.6201 | 2 |

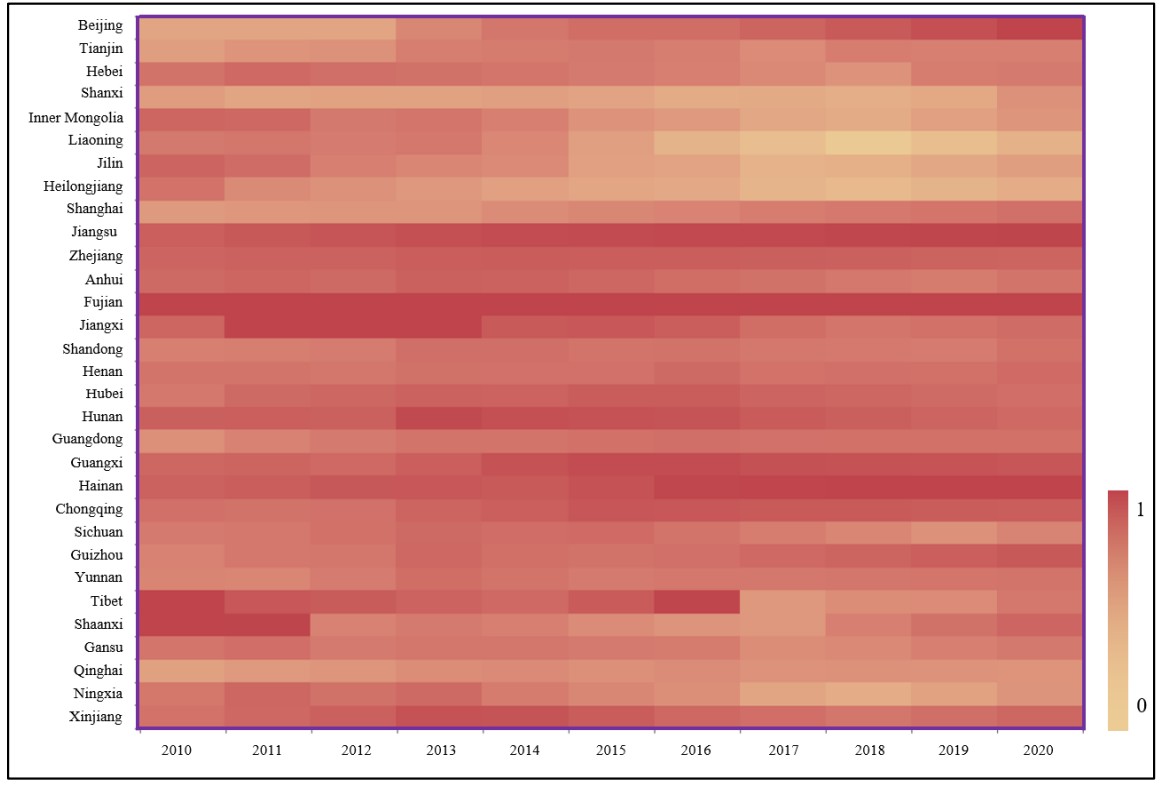

**Figure 5.** Heat map of the comparative production efficiencies of the construction industry in 31 provinces in China.

In this study, the integrated productivity values of the construction industry in each province from 2010 to 2020 are visualized to form a heat map. The productivity values of Jiangsu and Fujian have been stable at a high level in the last decade. Beijing, Hainan, Guangxi, Shanghai, and other provinces show a year-on-year upward trend of integrated production efficiency values. However, Hebei, Inner Mongolia, Jilin, Liaoning, Heilongjiang, and Jiangxi show a gradually decreasing trend. In addition, Shanxi, Heilongjiang, Qinghai, and other central and western cities have lower overall productivity values and no increasing trend.

### 4.5. Comparison of Regional Construction Industry Productivity Index Differences

In order to visually compare the static production efficiency value, dynamic efficiency value, and comprehensive production efficiency value of the three differences, a histogram of the three production efficiency values was constructed and is shown in Figure 6.

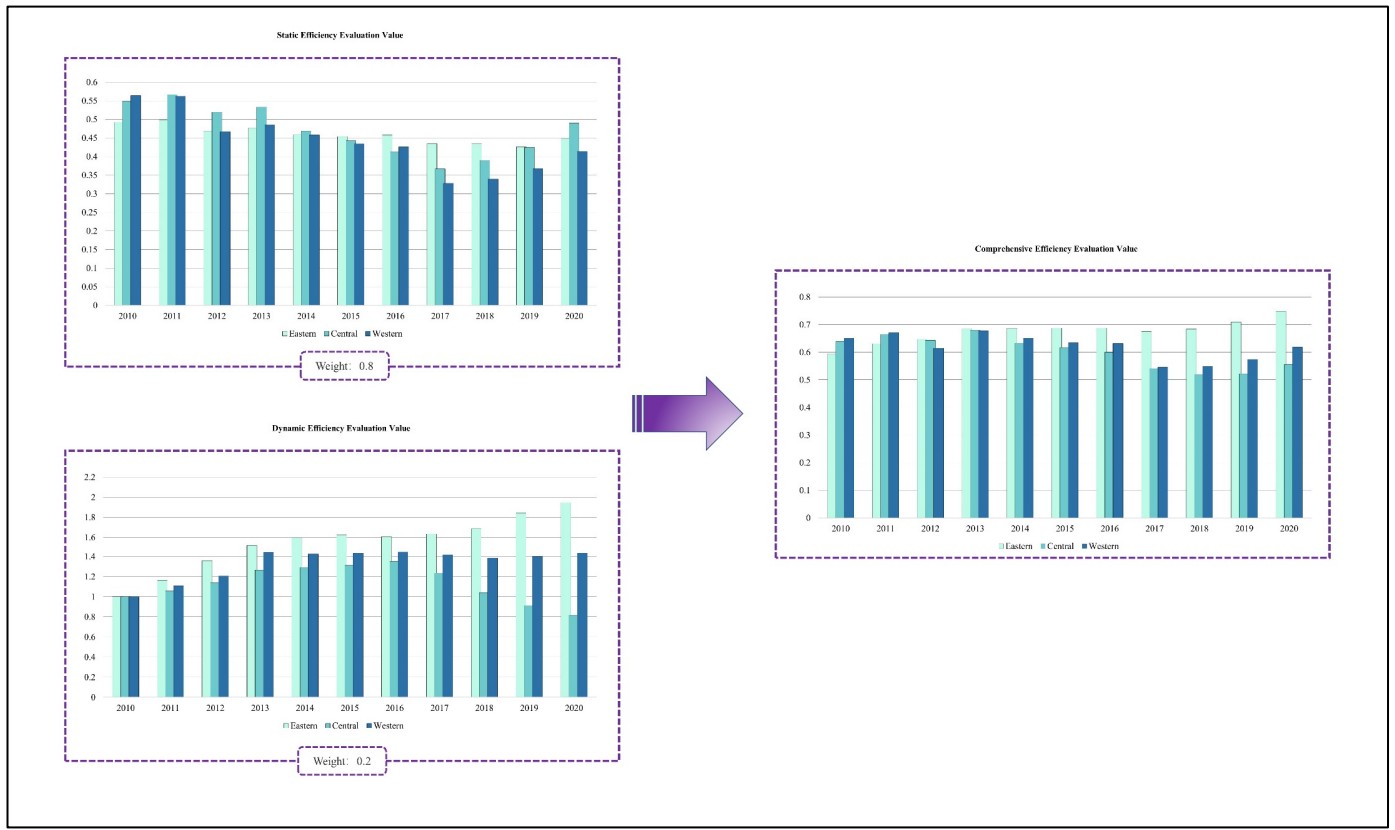

**Figure 6.** Histogram of construction productivity in the eastern, central, and western regions.

This study compares the different types of production efficiency values for the construction industry in the central, eastern, and western regions. As can be seen from the static production efficiency values, the three regions have alternately increased in production efficiency. The central region went from leading to ranking behind, while the western region has been stable in a more leading position. However, the overall view is that the static production efficiency has been at a lower level and the difference between the three regions' production efficiency values is small, i.e., the highest value is less than 0.6. From the dynamic production efficiency, it can be seen that the three regions of the construction industry production efficiency fluctuations have significantly increased. However, the opposite has been observed in regard to the static efficiency value measurement results and the western region dynamic efficiency. In contrast to the static efficiency value measurement results, the lowest dynamic efficiency value was found in the western region, which is between 1.0 and 1.8, while the dynamic efficiency value in the central region was found to be higher than those in the central region, and which shows a trend of growth and then a fluctuation toward a decrease; its value is between 1.0 and 2.2. It has been shown that the static bootstrap-DEA method mainly evaluates the resource allocation efficiency of a decision-making unit DMU, while the dynamic Malmquist-DEA method mainly evaluates the change in production efficiency. There are large differences in the input–output growth rate of each province due to different development levels in the base period. Therefore, relying only on the dynamic construction production efficiency values would overestimate the estimation results. From the integrated production efficiency values, the integrated evaluation results obtained by combining dynamic and static values can neutralize the

static and dynamic efficiency values, making the evaluation results more reasonable. From the graph, the central region and the western region are alternately leading year by year, and the production efficiency level of the central region starts to lead from 2017, while the eastern region lags behind the central and western regions from 2010 to 2013. However, the eastern region's value leads year by year from 2014, and its advantage thus increases year by year.

Overall, from the spatial dimension, the current static production efficiency values of China's construction industry vary less among the eastern, central, and western regions, ranging from 0.3 to 0.6. In addition, the eastern region does not show a production efficiency level that matches its economic level or the scale of the construction industry. From the dynamic production efficiency measurement results, the production efficiency of the construction industry from east to west shows a gradual development trend, with the highest in the eastern region, the second highest in the central region, and the lowest in the western region, thereby indicating a positive correlation between the degree of regional economic development and the dynamic production efficiency level of the construction industry. From the time series—except for the static production efficiency value, which fluctuates less—the dynamic production efficiency and comprehensive production efficiency show a steady increase. Moreover, the gap between the three regions is becoming bigger. The main reasons for this phenomenon may lie in the following four points: First, the marketization process of China's construction industry has been accelerating, and the strengthening of competition awareness has prompted enterprises to continuously improve their productivity. Second, the entry of numerous FDIs has brought advanced technology and management concepts, which have been diffused to Chinese enterprises through various ways and channels, thus providing a strong driving force for the improvement of the production and innovation efficiency of enterprises. Third, the increasing investment in science and technology innovation in the construction industry in recent years has greatly contributed to the optimization and upgrading of the current structure of China's construction industry. Fourth, China's regional economic development is extremely unbalanced, and the development of the construction industry shows large differences, leading to the optimal allocation of production factors and resources in the eastern region with good basic conditions, thereby showing the "Matthew effect" of production efficiency.

## 5. Discussion

Several interesting findings of this study must be highlighted. First, the selection and design perspectives of the efficiency evaluation indicators in the research are reasonable. The number of construction enterprises and the total assets of the construction industry are selected as input indicators through the value-added method and the Delphi method output indicators. This study provides a scientific and quantitative research method for constructing the construction industry's production efficiency evaluation index system. Second, the evaluation results of the static DEA model are evidently different before and after the correction, and the production efficiency level of the construction industry after the correction is evidently low. This fully shows that the original value of conventional DEA efficiency without bootstrap technology correction does overestimate the production efficiency of the construction industry. At the same time, compared with the efficiency value of the traditional DEA method, it is found that although the production efficiency value of the bootstrap-DEA method has a similar trend to the traditional DEA method, the bootstrap-DEA method is more robust than the traditional DEA method, and can more accurately reflect the construction industry in relevant provinces' productivity. Third, from the perspective of the static production efficiency value, the central region has ranked from leading to falling behind. Conversely, the western region has been stable in a relatively leading position. However, in general, the static production efficiency of China's construction industry has been at a low level, and the overall fluctuations are not large, but they have not reached an effective state. After considering the factors of the growth rate of the construction industry resource input and technological progress, the obtained dynamic pro-

duction efficiency value of the construction industry clearly magnifies the variation range of the construction industry production efficiency between the three regions. Additionally, it presents the distribution pattern of "eastern > central > western", and regional differences show a "gradual expansion" trend, in which technological progress is the main driving force for the improvement of production efficiency in China's construction industry. Finally, from the radar chart of the comprehensive production efficiency value, the comprehensive evaluation result of the combination of dynamic and static approaches can offset the static and dynamic efficiency values, thereby making the evaluation result more reasonable. It can be seen from the figure that the central and western regions alternately lead year by year, and the production efficiency level of the central region began to lead in 2017. However, overall, the comprehensive production efficiency of the construction industry in all regions is less than 1, which has not reached the effective state of resource allocation. In addition, there are significant differences between provinces, cities, and regions, and the coordinated development between regions should be strengthened.

## 6. Conclusions

The results of scientific and reasonable production efficiency evaluation are very relevant for the development of industrial policies [52]. However, previous studies have mainly used traditional DEA models to perform static efficiency or dynamic efficiency evaluation separately, which can reduce the accuracy of DEA model evaluation. Few studies have combined the two methods to examine the regional differences of construction production efficiency. This study combines four techniques—value added, Delphi, bootstrap DEA, and Malmquist index. To a certain extent, this research idea achieves an integrated study of the selection of production efficiency evaluation indicators and the establishment of evaluation systems and model metrics, which minimizes the influence of subjective factors and error factors, and provides a scientific quantitative evaluation method for the construction of a production efficiency evaluation index system. The main conclusions of this study are shown in the following: (1) The results of the comparison before and after the correction are obvious, and the production efficiency of the construction industry is significantly reduced after the correction. The resource allocation in the Chinese construction industry has not reached the optimal state, and obvious input redundancy exists in the enterprises. The output of certain regions is insufficient, and the difference in the regional development level is prominent; this can be seen in the static efficiency results all being less than 1. (2) The results of the Malmquist exponential decomposition show that the current trends of technical efficiency and scale efficiency in the construction industry are generally opposite to each other, indicating that the improvement of scale efficiency has not led to the improvement of management efficiency and the accumulation of production experience, which means that the construction industry's model of relying solely on scale expansion is no longer sustainable. More emphasis needs to be placed on technological innovation and investment in research and development. (3) The integrated efficiency score neutralizes the results of the static efficiency score and the dynamic efficiency score and can better reflect the development of productivity in the construction industry. The results of the integrated efficiency score show that the construction industry in the central and western regions has similar productivity levels, with the eastern region leading the construction industry in terms of productivity from 2014 onwards and with an increasing gap. It shows that the regional development of China's construction industry is uneven and needs to be adjusted with attention at the policy level. (4) This study proposes a set of ideas for evaluating the productivity of the construction industry, which covers the selection of indicators, model optimization, static and dynamic two-dimensional analysis, and comprehensive evaluation, and which can also be widely applied in other research areas.

This study has certain implications, which help provide references for policy makers. The obvious characteristics of regional differences require the government to formulate differentiated policies for different regions, pay more attention to regional integration development, prevent problems such as manufacturing policy depression and local protec-

tionism, and promote the coordinated development of the regional construction industry to a higher level and higher quality. Compared with previous studies, this study makes a number of theoretical and practical contributions. First, current research on assessing productivity in the construction industry tends to select indicator systems directly based on experience. In response to this weakness, this study completes the construction of an input–output indicator system using a combination of the value-added method and the Delphi method before conducting the evaluation of productivity in the construction industry, further ensuring the comprehensiveness and scientificity of the evaluation system. Second, previous studies have mainly focused on a single perspective of static performance or dynamic changes, which does not fully reflect the level of productivity development and potential of the construction industry, which has hindered the comprehensive evaluation of the productivity of the construction industry in different regions of China and the proposal of corresponding optimization strategies. The study uses the bootstrap-DEA and Malmquist index decomposition models to evaluate the static and dynamic productivity of the construction industry, respectively, and then arrives at the results of the comprehensive productivity evaluation of the construction industry, which is of great significance for the objective assessment of the current development situation of the construction industry.

This study also has certain limitations. First, the measurement of productivity indicators in the construction industry requires the consideration of complex and multifaceted factors. This study used a combination of subjective and objective methods to select indicators, but did not take into account the uncertainty of data collection and the impact that the type of data has on the combination of scores. How to avoid errors due to data uncertainty through model optimization deserves a more in-depth study. Second, this study provides a case study in a Chinese context to measure the combined productivity of the construction industry. While this study can be extended to other countries, the implementation needs to take into account the social and economic circumstances of other countries, and this needs to be further explored. It should be noted that this is an empirical study using only China as a case study, and that provincial-level data on China's construction industry were chosen due to limitations in data availability. In the future, we could try to conduct the study at a more micro level for prefecture-level cities. This study could also be improved by extending it to other countries or by using data from a longer period. In addition, this study analyzes differences in the productivity of the construction industry at the provincial level in China, but what factors contribute to these differences is also worthy of further in-depth study.

**Author Contributions:** Conceptualization, A.Y.; methodology, A.Y.; software, A.Y.; validation, A.Y.; formal analysis, X.Y.; writing—original draft preparation, X.Y.; writing—review and editing, A.Y.; visualization, A.Y.; supervision, X.Y.; and funding acquisition, X.Y. All authors have read and agreed to the published version of the manuscript.

**Funding:** This research was funded by the Junior Fellowships for Advanced Innovation Think-Tank Program of the China Association for Science and Technology (Grant No.: 2021ZZZLFZB1207134; Funder: Aobo Yue) and Henan Province Higher Education Teaching Reform Research and Practice Project (Grant No.: 2021SJGLX527; Funder: Xupeng Yin).

**Informed Consent Statement:** Not applicable.

**Data Availability Statement:** The data that support the findings of this study are openly available in the *China Statistical Yearbook* at http://www.stats.gov.cn/tjsj/ndsj/ (accessed on 10 December 2021).

**Acknowledgments:** The authors are very thankful to all the members of the research team and the editors whose invaluable comments and suggestions have helped to significantly improve the quality of this study.

**Conflicts of Interest:** The authors declare no conflict of interest.

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
