# Peer review of "Measuring Comprehensive Production Efficiency of the Chinese Construction Industry: A Bootstrap-DEA-Malmquist Approach"

_buildings, doi:10.3390/buildings13030834_

Round 1

Reviewer 1 Report

- Advantages and benefits of the proposed approach should be given in detail. Also, the research gaps and the novelty of this study should be highlighted and summarized.

- In the variables list of current study, some variables such as number of enterprises (Enterprise) and number of employees (Employees), are integer. Does the proposed approach have the ability to be used in the presence of integer values? For more details see the following references:

Kuosmanen, T., & Matin, R. K. (2009). Theory of integer-valued data envelopment analysis. European Journal of Operational Research, 192(2), 658-667.

Lozano, S., & Villa, G. (2006). Data envelopment analysis of integer-valued inputs and outputs. Computers & Operations Research, 33(10), 3004-3014.

- Generally, real data are tainted by uncertainty. The authors should discuss the proposed approach under data uncertainty.

- The quality of Figure 6 is not good.

- The authors should discuss on the generalization of the results of the study.

- The authors should fully explain all the details of their DEA model, such as returns to scale (RTS).

Reviewer 2 Report

This manuscript aims to propose a scientific and holistic framework to examine production efficiency in the construction industry. The topic is interesting, yet there are the following major concerns:

-        The contributions and research questions are not mentioned clearly in the introduction section.

-        The authors should mention other hybrid approaches to the DEA method. For example: Hybrid DEA-OPA, Hybrid DEA-AHP, etc.

-        The numbering system of the manuscript is problematic. For example: “1. Bootstrap-DEA Correction Measure Model”

-        The methodology section requires a major revision. The authors should rewrite it professionally.

-        There is an overlapping between the discussion and conclusion sections. Please modify them.

Reviewer 3 Report

Please carefully check the comments in the PDF version of the manuscript that has been attached. 

Round 2

Reviewer 1 Report

- Some future research directions should be suggested at the end of manuscript.

- Literature review and references should be updated according to recent studies (2022-2023).

Reviewer 2 Report

There is no more comments.

Author Response

Thank you for your support of this manuscript. On behalf of all the contributing authors, I would like to express our sincere appreciations of your letter and reviewers’constructive comments concerning our article entitled “Measuring Comprehensive Production Efficiency of the Chinese Construction Industry: A Bootstrap-DEA-Malmquist Approach” (Manuscript No: buildings-2153962).